# Evaluation of tuberculosis diagnostic test accuracy using Bayesian latent class analysis in the presence of conditional dependence between the diagnostic tests used in a community-based tuberculosis screening study

**Alfred Kipyegon Keter**[1,2,3]*, **Lutgarde Lynen**[1], **Alastair Van Heerden**[2,4], **Emily Wong**[5,6], **Klaus Reither**[7,8], **Els Goetghebeur**[3], **Bart K. M. Jacobs**[1]

**1** Department of Clinical Sciences, Institute of Tropical Medicine Antwerp, Antwerp, Belgium, **2** Centre for Community Based Research, Human Sciences Research Council, Pietermaritzburg, South Africa, **3** Department of Applied Mathematics, Computer Science and Statistics, Ghent University, Ghent, Belgium, **4** MRC/WITS Developmental Pathways for Health Research Unit, Department of Paediatrics, Faculty of Health Science, University of the Witwatersrand, Johannesburg, South Africa, **5** Africa Health Research Institute, Durban, South Africa, **6** Division of Infectious Diseases, Heersink School of Medicine, University of Alabama at Birmingham, Birmingham, AL, United States of America, **7** Swiss Tropical and Public Health Institute, Allschwil, Switzerland, **8** University of Basel, Basel, Switzerland

* aketer@ext.itg.be

**Data Availability Statement:** This work is based on analysis of secondary data published in the

## Abstract

Diagnostic accuracy studies in pulmonary tuberculosis (PTB) are complicated by the lack of a perfect reference standard. This limitation can be handled using latent class analysis (LCA), assuming independence between diagnostic test results conditional on the true unobserved PTB status. Test results could remain dependent, however, e.g. with diagnostic tests based on a similar biological basis. If ignored, this gives misleading inferences. Our secondary analysis of data collected during the first year (May 2018 –May 2019) of a community-based multi-morbidity screening program conducted in the rural uMkhanyakude district of KwaZulu Natal, South Africa, used Bayesian LCA. Residents of the catchment area, aged $\geq$15 years and eligible for microbiological testing, were analyzed. Probit regression methods for dependent binary data sequentially regressed each binary test outcome on other observed test results, measured covariates and the true unobserved PTB status. Unknown model parameters were assigned Gaussian priors to evaluate overall PTB prevalence and diagnostic accuracy of 6 tests used to screen for PTB: any TB symptom, radiologist conclusion, Computer Aided Detection for TB version 5 (CAD4TBv5$\geq$53), CAD4TBv6$\geq$53, Xpert Ultra (excluding trace) and culture. Before the application of our proposed model, we evaluated its performance using a previously published childhood pulmonary TB (CPTB) dataset. Standard LCA assuming conditional independence yielded an unrealistic prevalence estimate of 18.6% which was not resolved by accounting for conditional dependence among the true PTB cases only. Allowing, also, for conditional dependence among the true non-PTB cases produced a 1.1% plausible prevalence. After

articles referenced in the body of the paper. The data is freely available in the manuscripts cited. We obtained the second dataset from Africa Health Research Institute (AHRI) upon putting in a formal request where we entered a binding agreement to use the data solely for the intended research purpose. Data is publicly available but AHRI owns and controls access and use of the data. Upon request, the dataset can be accessed at the AHRI data repository (https://data.ahri.org/index.php/catalog/990/study-description#metadata-data_access).

**Funding:** This project is part of the European Union and Developing Countries Clinical Trials Partnership (EDCTP2) programme supported by the European Union (grant number: RIA2018D-2498; TB TRIAGE+), https://www.edctp.org/. LL, AVH, KR jointly applied for the EDCTP grant. The findings and conclusions of this work are guaranteed by the authors and do not necessarily represent the official position of the funders. The funders had no role in the study design, data collection and analysis, decision to publish, or preparation of the manuscript.

**Competing interests:** The authors have declared that no competing interests exist.

incorporating age, sex, and HIV status in the analysis, we obtained 0.9% (95% CrI: 0.6, 1.3) overall prevalence. Males had higher PTB prevalence compared to females (1.2% vs. 0.8%). Similarly, HIV+ had a higher PTB prevalence compared to HIV- (1.3% vs. 0.8%). The overall sensitivity for Xpert Ultra (excluding trace) and culture were 62.2% (95% CrI: 48.7, 74.4) and 75.9% (95% CrI: 61.9, 89.2), respectively. Any chest X-ray abnormality, CAD4TBv5≥53 and CAD4TBv6≥53 had similar overall sensitivity. Up to 73.3% (95% CrI: 61.4, 83.4) of all true PTB cases did not report TB symptoms. Our flexible modelling approach yields plausible, easy-to-interpret estimates of sensitivity, specificity and PTB prevalence under more realistic assumptions. Failure to fully account for diagnostic test dependence can yield misleading inferences.

# Introduction

The World Health Organization (WHO) recommends that a case of tuberculosis (TB) be considered "bacteriologically confirmed" if the biological specimen yields a positive test result on smear microscopy, culture, or WHO-recommended rapid molecular test for TB (e.g. Xpert MTB/RIF) [1, 2]. A combination of any two or all of the three microbiological tests can also be used to ascertain the presence of active TB. However, inference based on an imperfect reference standard, or a combination of imperfect reference standards, is prone to bias. To overcome this problem, latent class analysis (LCA) can be used to evaluate imperfect diagnostic tests, including the reference standard itself. This approach is used for identifying unobserved mutually exclusive subgroups in the population using information from the measured individual characteristics [3]. It has enjoyed extensive application in many disciplines, medical and non-medical. In the medical field, extensive applications have been carried out in HIV and cancer screening [4, 5]. Over the past few decades, there has been growing interest in the biomedical field, including the evaluation of diagnostic tests in the absence of a gold standard in the field of infectious diseases, in both human and veterinary medicine [4, 6]. LCA was applied in the early 1980s to evaluate the performance of two skin tests, Tine and Mantoux, for the detection of TB [7, 8]. Recently, it was applied to evaluate the diagnostic tests used to diagnose children suspected of childhood pulmonary TB (CPTB) in South Africa [9]. In its application, standard LCA assumes that conditional on the true disease status the diagnostic test misclassification error rates (MER) are independent. This implies that the MER of the diagnostic tests are also constant across the underlying subpopulations [7]. This conditional independence model (CIM), yields misleading inferences when the data under consideration violate the assumptions. In practice, diagnostic test dependence can remain even after conditioning on the true disease status, especially for diagnostic tests based on a similar biological basis.

For example, the three sputum-based microbiological tests for detecting mycobacterium bacilli are more likely to be dependent among the true PTB cases because individuals with higher bacillary load are more likely to test positive for active TB on the three diagnostic tests [9, 10]. This dependence is likely higher when the same sputum sample is used. Conversely, the absence of TB bacilli in the sample induces negligible dependence. Hence, conditional on the true non-TB status the three microbiological tests are presumed independent [9, 10]. TB-compatible symptoms such as cough, fever, night sweats, and unexplained weight loss among others are manifestations of the severity of the disease encoded in high bacillary load. However, a high proportion of bacteriologically confirmed cases are asymptomatic [11]. Similarly, radiological methods such as radiologist interpretation of chest X-ray and Computer Aided

Detection for TB (CAD4TB) are more likely to be positive when true TB patients are symptomatic because positivity rates are linked to the progression of the disease and increasing damage to the lungs. Nonetheless, a high proportion of bacteriologically confirmed cases with chest X-ray abnormalities are also asymptomatic [12]. This reveals a low chance of dependence between (any) TB symptom, radiological methods and microbiological tests among the true PTB cases. On the contrary, radiologist interpretation of chest X-ray images and CAD4TB are more likely dependent among true PTB cases.

TB symptoms can also be associated with other etiologies such as bacterial pneumonia, malaria, non-tuberculous mycobacteria and chronic obstructive pulmonary disease (COPD). Hence, they can also be wrongly interpreted as suggestive of PTB by a clinician, a radiologist or CAD4TB that identifies abnormalities in the chest using chest X-ray images. This also induces dependence between TB symptoms, radiologist conclusion and CAD4TB among the true non-PTB cases. Additionally, a reported history of recent TB infection can be identified by chest X-ray and Xpert MTB/RIF. This also creates a conditional dependence between X-ray-based methods and Xpert MTB/RIF among the true non-PTB cases.

Further, the performance of TB symptom screening, the three microbiological tests and radiography-based diagnosis can also be affected by comorbidities such as advanced HIV disease. This induces additional dependence between these diagnostic methods, separately among the true PTB cases and the true non-PTB cases [9]. Hence, the estimation of PTB prevalence and properties of TB symptoms, microbiological tests and radiography-based diagnosis in the presence of test dependence is nontrivial. Using standard CIM is therefore expected to yield incorrect inferences.

In a quest to circumvent the limitations of CIM with 2 latent classes, models that allow isolation of the covariance between diagnostic tests have been proposed [8, 10, 13]. Dendukuri and Joseph proposed Bayesian versions of the fixed effects model by Vacek (1985) and the random effects model by Qu et al (1996) [14]. Although the random effects LCA could handle higher-order covariance terms, it proceeds under distributional assumptions and functional relationships between the diagnostic test properties and the random effects that may not be verifiable from the data. While arguing that modelling of covariance terms is difficult to combine with expert opinion, Berkvens et al (2006) extended the approach based on conditional probabilities [15] that was previously applied to two diagnostic tests [16]. We find this method the most attractive and appealing because it is easy to relate to clinical practice and easy to elicit prior distributions for the parameters from experts. This approach involves the specification of the model for the joint probability of a combination of multiple diagnostic tests as a function of the overall disease prevalence, sensitivity and specificity of the first diagnostic test and conditional probabilities of the other diagnostic tests using the chain rule of conditional probability. We propose a modification to this approach such that the conditional probabilities of the diagnostic tests are modelled using regression methods for dependent binary outcomes [17]. The probit link function is used to relate the outcome and the independent variables [18]. Hence, we call these models *regressive probit models*. We extend the method to incorporate measured categorical variables known to affect the prevalence and diagnostic test accuracy [19]. Higher-order covariance terms can be estimated by incorporating higher-order interaction terms in the model. Priors will then be assigned to the regression parameters. Though not a requirement, this approach benefits from some logical ordering in the diagnostic tests. This aligns well with algorithms where a subject usually undergoes initial screening using a WHO TB symptom screen followed by one or more diagnostic tests, culminating in confirmatory testing using Xpert MTB/RIF or culture.

We evaluated our proposed approach using previously published data of hospitalized children suspected of childhood pulmonary TB (CPTB) in two facilities in Cape Town, South

Africa [9]. We then applied our method to analyze data from a community-based multi-mor-bidity survey conducted in a rural district of KwaZulu Natal province, South Africa [20].

## Methods

### Model

Let the random variable $Y_j, j = 1,2,...,J$ denote the $j^{th}$ diagnostic test and the random variable $D$ denote the latent (unobserved/unmeasured) disease status such that $Y_j = 0(1)$ if the $j^{th}$ diagnostic test result is negative (positive) and $D = 0(1)$ if the true disease status is negative (positive). Under the assumption of conditional independence, the joint probability of a combination of test results from a set of $J$ diagnostic tests $\boldsymbol{Y} = (Y_1, Y_2, \cdots, Y_J)$ is given by [21]

$$Pr(\boldsymbol{y}) = \sum_{d=0}^{1} Pr(D = d) \prod_{j=1}^{J} Pr(Y_j = y_j | D = d) \tag{1}$$

However, for $J$ dependent diagnostic tests Eq (1) is wrong and will lead to incorrect inferences. Under Bayesian inference, we extended Eq (1) to allow modelling of the joint probability of a set of $J$ dependent diagnostic tests in a function of disease prevalence, sensitivity and specificity of the first diagnostic test and such probabilities of the earlier diagnostic tests conditional on the results of already modelled test outcomes, using chain-rule of conditional probability (Section 1 in S1 File). We extended the model to additionally include observed covariates known to affect the diagnostic accuracy and/or prevalence.

### Applications

**Childhood pulmonary tb (cptb) data.**   This study was conducted between February 2009 and June 2014 in two hospitals in Cape Town, South Africa among hospitalized children suspected of CPTB. The participants were consecutively enrolled if they presented with signs and symptoms akin to CPTB. Children aged below 15 years with cough lasting >2 weeks were enrolled if they had a household contact with TB in the past 3 months preceding enrollment, lost or failed to gain weight in the past 3 months preceding enrollment, had a positive skin test to purified protein derivative, or had a chest radiography suggestive of CPTB. Children who had received TB treatment or prophylaxis for >72 hours before enrollment, who were not residents of Cape Town, who failed to produce induced sputum specimens or whom the legal parent or guardian failed to give informed consent were excluded. The diagnostic tests involved in this study were 1) tuberculin skin test (TST), 2) chest X-ray (radiography), 3) smear microscopy, 4) Xpert MTB/RIF (Xpert) and 5) culture. Further details of the study are found elsewhere [9]. According to the experts in the original study, the three microbiological tests based on sputum samples (smear microscopy, Xpert and culture) are more likely dependent in children with CPTB because the sensitivities of these tests are functions of the severity of CPTB [9]. Chest X-ray findings were assumed to have negligible dependence with the other diagnostic tests because it is based on a different marker of CPTB. TST was assumed to be negatively correlated with microbiological tests among children with severe disease because TST has been reported to be less sensitive in persons, particularly adults, with a severe disease which can be associated with a higher bacillary burden. All these diagnostic tests were assumed to be independent among the children without CPTB [9]. In the previous analysis, the authors used a Gaussian random effect on the probit scale to represent CPTB bacterial load. They also allowed the sensitivities of culture, Xpert and smear microscopy and the sensitivity of TST to, respectively, have linear and quadratic functions of this random effect. The sensitivity of the chest X-ray was held constant. We aimed to evaluate evidence of residual dependence amongst

test errors and its impact on the estimated prevalence and diagnostic test properties, and compare our findings to those obtained in the previous two studies [9, 10]. Data for a total of 749 children were included in our analysis.

**Active TB case-finding study in KwaZulu-Natal, South Africa ("Vukuzazi" study).** The data for this analysis were collected in the rural uMkhanyakude district of northern KwaZulu Natal during the first year (May 2018—May 2019) of a community-based multimorbidity screening programme in South Africa ("Vukuzazi" study) [20, 22, 23]. This dataset was accessed and analyzed in conformity with the binding agreement contained in [20]. Individuals aged ≥15 years and residents of households in the Africa Health Research Institute (AHRI) Data Surveillance System (DSS) area were eligible for enrollment. Briefly, posterior-anterior digital chest X-rays were obtained from the participants and subsequently interpreted by CAD4TB version 5 (CAD4TBv5). CAD4TBv5 calculated a score (ranging from 0–100) indicating lung abnormalities and the likelihood of active pulmonary TB. Within seven days of enrollment, an expert radiologist with >35 years of experience reviewed all the chest X-rays in a central setting blinded from CAD4TBv5 score and any other patient information and classified them as depicting either normal or abnormal lung fields. For the chest X-ray images depicting abnormal lung fields, the radiologist further classified them as highly suggestive of active TB or not suggestive of active TB. CAD4TBv6, an updated version of CAD4TBv5, became available after data collection and was used to calculate CAD4TBv6 scores retrospectively. In total, 9914 participants meeting the eligibility criteria were enrolled in the study. Following WHO guidelines for TB prevalence surveys, participants were eligible for microbiological testing if they reported any cardinal TB symptom (fever, night sweats, weight loss or cough) or if they had an abnormal chest X-ray as indicated by CAD4TBv5 score above a predefined triaging threshold or if the radiologist indicated abnormal lung fields despite CAD4TBv5 score below the threshold. Sputum samples from 4976 participants were analyzed for *Mycobacterium tuberculosis* using Xpert Ultra MTB/RIF® (Xpert Ultra) and liquid MGIT culture. Further details of the study are found elsewhere [12, 22, 23]. In this analysis, we consider any TB symptom ($Y_1$), radiologist conclusion ($Y_2$), CAD4TBv5≥53 ($Y_3$), CAD4TBv6≥53 ($Y_4$), Xpert Ultra ($Y_5$), and culture ($Y_6$) (Fig 1). A technical description of how the thresholds for CAD4TB version 5 and version 6 were determined is given in Section 3 of the S1 File. Xpert Ultra, when positive, provides a semiquantitative result defined either as very low, low, medium or high. It also provides an additional category called "trace" classified as the lowest level in the semiquantitative scale. It has low quantities of Deoxyribonucleic Acid (DNA) and often corresponds to paucibacillary specimens [12, 24, 25]. Due to the emerging uncertainty regarding the clinical significance of "trace" laboratory findings that occur as the only microbiological evidence for TB [12, 26], we excluded this category from the Xpert Ultra positive i.e. it is defined as negative (hereinafter defined as Xpert Ultra (excluding trace)). A total of 4976 participants eligible and tested with microbiological tests were included in our analysis.

As depicted in Fig 1, the TB bacillary load ($U_1^+$), which indicates the presence of PTB and is the source of conditional dependence between microbiological tests, is usually not routinely measured (or observed). Similarly, the radiological features ($U_2^+$) such as cavitation, volume loss and fibrosis in the lungs among other manifestations that indicate the extent of lung damage are also not usually measured. The presence of these radiological changes ($U_2^+$) among the symptomatic individuals with PTB induces dependence between radiological interpretations and CAD4TB results. Given that higher bacillary load is linked to PTB-compatible symptoms such as cough which is a consequence of cavitation in the lungs attributable to higher bacillary load, the unmeasured radiological features ($U_2^+$) indicative of the extent of lung damage is correlated with the unmeasured (high) bacillary load ($U_1^+$). The unmeasured radiological features

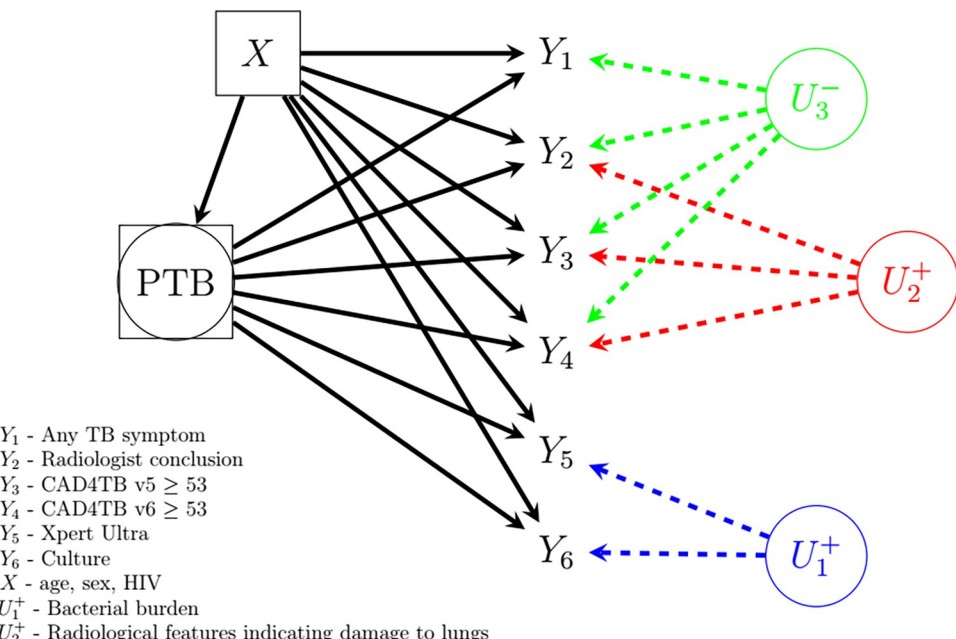

**Fig 1. Heuristic model of diagnostic tests, unobserved PTB status, measured covariates and unobserved sources of diagnostic test dependence for Vukuzazi data.**

$(U_3^-)$ due to other etiologies such as bacterial pneumonia, past PTB and COPD among symptomatic individuals without PTB induces dependence between any TB symptom, radiological interpretations and CAD4TB results.

## Statistical analysis

In the analysis of CPTB data, we allowed TST ($Y_1$), radiography ($Y_2$), microscopy ($Y_3$), Xpert ($Y_4$) and culture ($Y_5$) as the possible ordering of the diagnostic tests. First, we fitted the conditional independence model (CIM) (Model 0). Next, we hierarchically fitted three other models allowing for conditional dependence between the diagnostic tests. The models considered in the analysis of CPTB data and the dependencies allowed are presented in Table 1. Model 1 was based on expert opinion [9], Model 2 adds dependence between TST and radiography in non-CPTB cases to Model 1, while Model 3 additionally allows dependency of all diagnostic tests with radiography among the true CPTB cases. Model 4 is the reduced version of Model 3 that omits the dependence between TST and radiography in non-CPTB cases. The structure of the models and the priors assigned to the parameters are presented in (S1 Table in S1 File).

For Vukuzazi, we allowed any TB symptom ($Y_1$), radiologist conclusion ($Y_2$), CAD4TBv5$\geq$53 ($Y_3$), CAD4TBv6$\geq$53 ($Y_4$), Xpert Ultra (excluding trace) ($Y_5$) and culture ($Y_6$) as the sequence of the available set diagnostic indicators. Using the radiologist's conclusion based on the chest X-ray findings we derived two variables: any chest X-ray abnormality (vs. normal lung fields), and chest X-ray abnormality suggestive of active TB (vs. abnormalities NOT suggestive of active TB combined with normal lung fields). We then performed two sets of analyses to evaluate the set of six diagnostic tests. The first analysis incorporates the radiologist's conclusion as any chest X-ray abnormality in the set of six diagnostic tests. The second incorporates the radiologist's conclusion as chest X-ray abnormality suggestive of active TB in the set of six diagnostic tests.

**Table 1. Models considered in the analysis of CPTB and Vukuzazi datasets.**

| CPTB | Dependencies allowed | |
|---|---|---|
| | **TB-positive** | **TB-Negative** |
| Model 0 | • None | • None |
| Model 1 | • Between TST ($Y_1$), microscopy ($Y_3$), Xpert ($Y_4$) and culture ($Y_5$) | • None |
| Model 2 | • Between TST ($Y_1$), microscopy ($Y_3$), Xpert ($Y_4$) and culture ($Y_5$) | • TST ($Y_1$) and radiography ($Y_2$) |
| Model 3 | Between TST ($Y_1$), radiography ($Y_2$), microscopy ($Y_3$), Xpert ($Y_4$) and culture ($Y_5$) | • TST ($Y_1$) and radiography ($Y_2$) |
| Model 4 | • Between TST ($Y_1$), radiography ($Y_2$), microscopy ($Y_3$), Xpert ($Y_4$) and culture ($Y_5$) | • None |
| **Vukuzazi** | | |
| Model 0 | • None | • None |
| Model 1 | • Between radiologist conclusion ($Y_2$), CAD4TBv5$\geq$53 ($Y_3$) and CAD4TBv6$\geq$53 ($Y_4$)<br>• Between Xpert Ultra ($Y_5$) and culture ($Y_6$) | • None |
| Model 2 | • Between radiologist conclusion ($Y_2$), CAD4TBv5$\geq$53 ($Y_3$) and CAD4TBv6$\geq$53($Y_4$)<br>• Between Xpert Ultra ($Y_5$) and culture ($Y_6$) | • Any TB symptom ($Y_1$), radiologist conclusion ($Y_2$), CAD4TBv5$\geq$53 ($Y_3$), CAD4TBv6$\geq$53 ($Y_4$) |
| Model 3 | • Between radiologist conclusion ($Y_2$), CAD4TBv5$\geq$53 ($Y_3$), CAD4TBv6$\geq$53 ($Y_4$), Xpert Ultra ($Y_5$) and culture ($Y_6$) | • Any TB symptom ($Y_1$), radiologist conclusion ($Y_2$), CAD4TBv5$\geq$53 ($Y_3$), CAD4TBv6$\geq$53 ($Y_4$) |
| Model 4 | • Between any TB symptom ($Y_1$), radiologist conclusion ($Y_2$), CAD4TBv5$\geq$53 ($Y_3$), CAD4TBv6$\geq$53 ($Y_4$), Xpert Ultra ($Y_5$) and culture ($Y_6$) | • Any TB symptom ($Y_1$), radiologist conclusion ($Y_2$), CAD4TBv5$\geq$53 ($Y_3$), CAD4TBv6$\geq$53 ($Y_4$) |

First, we fitted the CIM (Model 0). Next, we hierarchically fitted four other models allowing for conditional dependence between the diagnostic tests. Similarly, the models considered in the analysis of Vukuzazi data and the dependencies allowed are presented in Table 1. Model 1 relaxes the assumptions in Model 0 to allow conditional dependence between the radiologist conclusion, CAD4TBv5 and CAD4TBv6 and conditional dependence between Xpert Ultra (excluding trace) and culture among the true PTB cases. Model 2 adds to Model 1 the conditional dependence between any TB symptom, radiologist conclusion, CAD4TBv5 and CAD4TBv6 among the true non-PTB cases. Model 3 relaxes the assumptions in Model 2 to additionally allow conditional dependence between all the diagnostic tests except any TB symptom among the true PTB cases. Model 4 further relaxes the assumptions in Model 3 to additionally allow conditional dependence between all the diagnostic tests among the true PTB cases. Model 2 was identified as the model that best explains the conditional dependence in the data. Consequently, the conditional probabilities in the model were adjusted for the available measured covariates (age, sex and HIV status) known to affect the performance of the diagnostic tests and the prevalence of PTB. This model is presented in Fig 1. There was no evidence of varying estimates of specificity for Xpert Ultra (excluding trace) and culture across age, sex and HIV subpopulations. Hence, the analysis did not adjust for these covariates in the corresponding probit regression models. From the available dataset in the repository, age was already categorized in a ten-year interval. However, in our analysis, we collapsed the age groups into twenty-year intervals to avoid overfitting among the true PTB cases. Model selection was based on deviance [27, 28]. Among the competing models, the model with smaller deviance is considered to fit the data well. We also calculated the root mean square error (RMSE) deviation, defined as $\sqrt{E(\hat{f} - f)^2}$ where $\hat{f}$ and $f$ are the predicted and observed frequencies respectively, to assess the predictive power of the model. A smaller estimate of RMSE indicates good predictive power of the model among the competing models. Stochastic modelling using LCA was used to determine the threshold scores for CAD4TBv5 and CAD4TBv6. A detailed description and illustration of how the cut-off values were determined are given in (Section 3 in S1 File). The plausibility for the inclusion of CAD4TBv5 and CAD4TBv6 in the same model was assessed using a scatter plot of the scores for the two versions (S3 Fig in

*S1 File*). Using Model 2 adjusted for age, sex and HIV status, we provide a sensitivity analysis with the Xpert Ultra positive including the "trace" category (S15 Table in *S1 File*).

Using probit regression methods for dependent binary outcomes, we sequentially regressed each binary test outcome on the other observed test results conditional on the true unknown PTB status. Model parameters were assigned Gaussian priors (S1-S5 Tables in S1 File). Expert opinion was utilized to understand the potential dependencies and prior knowledge of some parameters. Inferences were based on the median summaries of the posterior distributions of PTB prevalence, diagnostic test sensitivity and specificity. We present the estimates and the 95% credible intervals (95% CrI).

We ran 50,000 Monte Carlo iterations with the first 25,000 discarded as 'burn-in'. For all analyses, convergence in model fitting was assessed by running three parallel chains. To reduce autocorrelation every 10th iteration was saved ("thinning") [29]. Trace plots and Gelman-Rubin convergence diagnostics were used to monitor mixing in the chains (S4-S10 Figs in S1 File). All analyses were implemented in R version 4.2.1 using R2jags package for R version 4.2.1. [30, 31]

## Inclusivity in global research

Additional information regarding the ethical, cultural, and scientific considerations specific to inclusivity in global research is included in the S1 Checklist.

# Results

## Childhood pulmonary TB (CPTB) data

Results from the analysis of CPTB data are shown in Table 2. The results from the model under the assumption of conditional independence (Model 0) can be misleading. Specifically, this model overestimates the sensitivity of culture thus incorrectly portraying this diagnostic test as (near) perfect. The model based on expert opinion (Model 1) gives similar results to those reported in a previous study [10]. The results reported in [9] are slightly different from our findings possibly because the model was adjusted for measured covariates. However, the 95% credible intervals for the estimates from all the models overlap substantially. Based on the deviance, the model that accounts for conditional dependence between all the diagnostic tests except radiography among children with CPTB and between TST and radiography among the children without CPTB (Model 2) performed as good as the model that accounts for conditional dependence between all the diagnostic tests among children with CPTB and between TST and radiography among the children without CPTB (Model 3). Nonetheless, the reduction in deviance from 102.1 for Model 2 to 101.7 for Model 3 does not outweigh the complexity in Model 3. This leaves Model 2 as the model of choice. Model 2 produced estimates of sensitivity for microbiological tests (smear microscopy, Xpert and culture) that are higher and the estimates of sensitivity and specificity for TST and radiography that are lower than those based on the expert opinion as given by Model 1 as well as those reported in [10].

The apparent difference between the results based on the expert opinion and those based on Model 2 explains the ability of Model 2 to account for the overlooked dependence between TST and radiography. Thus, Model 2 produces plausible estimates of prevalence and diagnostic test sensitivity and specificity.

## Vukuzazi data

This dataset comprised a total of 9914 participants. Of these participants, 4976 individuals were tested using Xpert Ultra and culture. Of these microbiologically tested participants, 1818

**Table 2. Posterior median and 95% credible intervals (95% CrI) of the prevalence and diagnostic test sensitivity and specificity of the CPTB dataset.**

| Test | Parameter | Model 0 | Model 1 | Model 2 | Model 3 | [a] Schumacher et al. [9] | [b] Wang et al. [10] |
|---|---|---|---|---|---|---|---|
| | | Median (95% CrI) | Median (95% CrI) | Median (95% CrI) | Median (95% CrI) | Median (95% CrI) | Median (95% CrI) |
| | Prevalence | 16.3 (13.6, 19.3) | 21.6 (15.6, 28.6) | 18.4 (14.7, 26.4) | 18.9 (15.1, 27.0) | 26.7 (20.8, 35.2) | 22.7 (16.7, 31.6) |
| TST | Sensitivity | 69.2 (60.5, 76.9) | 73.2 (61.8, 82.4) | 68.3 (58.5, 79.9) | 69.5 (59.4, 81.5) | 75.2 (61.2, 83.8) | 70.7 (59.9, 79.8) |
| | Specificity | 62.3 (58.4, 66.1) | 65.2 (59.6, 71.8) | 62.7 (58.5, 69.6) | 63.1 (58.8, 70.4) | 69.3 (63.2, 75.9) | 65.9 (60.3, 72.6) |
| Rad. | Sensitivity | 66.0 (57.1, 74.1) | 65.1 (56.1, 73.8) | 64.5 (55.5, 73.3) | 60.1 (42.5, 72.2) | 64.2 (54.9, 72.8) | 64.7 (55.9, 73.3) |
| | Specificity | 73.1 (69.5, 76.5) | 74.8 (70.5, 79.2) | 73.2 (69.4, 77.9) | 72.2 (67.8, 76.5) | 78.0 (73.4, 83.4) | 76.0 (71.3, 81.5) |
| SMM | Sensitivity | 33.8 (25.5, 42.5) | 24.3 (16.7, 34.8) | 28.4 (18.7, 37.6) | 28.4 (19.0, 37.2) | 22.3 (15.6, 30.3) | 26.7 (18.2, 37.6) |
| | Specificity | 99.9 (99.5, 100) | 99.9 (99.5, 100) | 99.9 (99.5, 100) | 99.9 (99.5, 100) | 99.7 (99.0, 100.0) | 100 (99.5, 100) |
| Xpert | Sensitivity | 74.8 (66.3, 82.5) | 59.1 (44.2, 75.7) | 69.4 (47.8, 78.4) | 70.3 (47.9, 79.5) | 49.4 (37.7, 62.2) | 57.2 (42.0, 73.1) |
| | Specificity | 98.2 (97.0, 99.3) | 98.7 (97.1, 99.9) | 98.7 (97.2, 99.9) | 99.2 (97.5, 100) | 98.6 (97.3, 99.5) | 98.8 (97.3, 100) |
| Culture | Sensitivity | 99.9 (91.0, 100) | 75.3 (56.9, 97.2) | 88.0 (61.8, 98.4) | 86.4 (60.3, 98.5) | 60.0 (45.7, 75.5) | 68.9 (50.7, 87.5) |
| | Specificity | 99.8 (99.0, 100) | 99.8 (99.2, 100) | 99.8 (99.2, 100) | 99.8 (99.2, 100) | 99.6 (98.7, 100.0) | 99.7 (98.6, 100) |
| | Deviance | 183.7 | 104.7 | 102.1 | 101.7 | - | - |
| | RMSE | 42.4 | 38.0 | 12.0 | 9.1 | 8.4 | 27.5 |

Model 0 –Based on the assumption of conditional independence

Model 1 –Based on the expert opinion as detailed in [9]. The model accounts for conditional dependence between all the diagnostic tests except radiography among children with CPTB and conditional independence between all the diagnostic tests among children without CPTB

Model 2 –The model accounts for conditional dependence between all the diagnostic tests except radiography among children with CPTB and conditional dependence between TST and radiography among children without CPTB

Model 3 –Accounts for conditional dependence between all the diagnostic tests among children with CPTB and conditional dependence between TST and radiography among children without CPTB

TST–Tuberculin skin test

Rad.–Radiography

SMM–Sputum smear microscopy

Xpert–Xpert MTB/RIF

RMSE–Root mean square error. This is calculated as the square root of the sum of squared differences between the observed frequencies and the predicted frequencies. It shows how good the model is in explaining the variability in the data (smaller is better)

[a] Random effects model with a dependence structure based on expert opinion used in Model 1 (adjusted for age, HIV, malnutrition and household TB contact)

[b] Fixed effects model with a dependence structure based on expert opinion used in Model 1

CrI–Credible Intervals

Note: The estimates presented in the table are percentages

(36.5%) were males and 1496 (30.2%) were HIV+ (15 subjects missing HIV status). Based on age distribution, 1158 (23.3%), 1295 (26.0%), 1728 (34.7%) and 794 (16.0%) were aged 15–29, 30–49, 50–69 and ≥70 years respectively (1 missing age). There were 849 (17.1%) individuals with any TB symptom, 1693 (34.0%) with any chest X-ray abnormality, 177 (3.6%) with chest X-ray abnormality suggestive of active TB, 1028 (20.7%) with CAD4TBv5≥53, 898 (18.0%) with CAD4TBv6≥53, 48 (1.0%) Xpert Ultra (excluding trace) positive test results and 55 (1.1%) with culture positive test results. Bacteriologically confirmed cases (Xpert Ultra (excluding trace) positive or culture positive) were 75 (1.5%). After excluding the individuals with missing data on age and HIV status, 4960 participants of which 74 were bacteriologically confirmed TB cases were available for analysis. We evaluated the diagnostic accuracy of any TB symptom, any chest X-ray abnormality, CAD4TBv5≥53, CAD4TBv6≥53, Xpert Ultra (excluding trace) and culture.

Using Bayesian LCA, we fit five different models on the subset tested using Xpert Ultra and culture. Table 3 shows that the model under conditional independence assumption (Model 0) leads to unrealistic estimates and incorrect inference. The model that considered conditional

**Table 3. Posterior median and 95% credible intervals (95% CrI) of the prevalence and diagnostic test sensitivity and specificity of Vukuzazi dataset.**

| Test | Parameter | Model 0 | Model 1 | Model 2 | Model 3 | Model 4 |
|---|---|---|---|---|---|---|
| | | Median (95% CrI) | Median (95% CrI) | Median (95% CrI) | Median (95% CrI) | Median (95% CrI) |
| | Prevalence | 18.6 (17.4, 19.8) | 20.5 (18.9, 22.4) | 1.1 (0.8, 1.5) | 1.2 (0.8, 1.6) | 1.2 (0.8, 1.6) |
| Any TB | Sensitivity | 20.5 (18.0, 23.3) | 20.2 (17.7, 22.8) | 27.5 (17.5, 39.7) | 27.0 (17.1, 38.6) | 27.8 (17.5, 40.1) |
| symptom | Specificity | 83.6 (82.5, 84.8) | 83.6 (82.4, 84.8) | 83.0 (81.9, 84.0) | 83.0 (81.9, 84.0) | 83.0 (81.9, 84.0) |
| Radiologist | Sensitivity | 90.4 (88.2, 92.5) | 87.2 (82.5, 90.5) | 83.1 (72.5, 90.9) | 83.1 (72.1, 91.1) | 83.7 (72.9, 91.5) |
| conclusion‡ | Specificity | 79.3 (78.0, 80.6) | 80.1 (78.6, 81.6) | 66.8 (65.4, 68.1) | 66.8 (65.4, 68.1) | 66.8 (65.4, 68.1) |
| CAD4TBv5≥53 | Sensitivity | 84.6 (81.7, 87.3) | 79.4 (74.0, 83.7) | 79.1 (67.1, 88.4) | 78.9 (67.0, 88.2) | 79.4 (67.3, 88.8) |
| | Specificity | 94.4 (93.5, 95.2) | 94.8 (93.9, 95.8) | 80.1 (79.0, 81.2) | 80.2 (79.0, 81.2) | 80.2 (79.0, 81.2) |
| CAD4TBv6≥53 | Sensitivity | 88.2 (85.1, 91.1) | 83.0 (78.1, 87.1) | 79.7 (68.0, 88.9) | 79.9 (68.0, 89.1) | 79.7 (67.2, 89.0) |
| | Specificity | 98.4 (97.8, 98.9) | 98.9 (98.2, 100) | 82.6 (81.4, 83.6) | 82.6 (81.5, 83.6) | 82.6 (81.5, 83.7) |
| Xpert Ultra† | Sensitivity | 11.7 (9.9, 13.7) | 5.1 (3.8, 6.6) | 60.0 (46.9, 72.6) | 59.7 (45.5, 72.9) | 60.1 (45.5, 73.9) |
| | Specificity | 100 (99.9, 100) | 99.4 (99.2, 99.6) | 99.3 (99.1, 99.5) | 99.4 (99.1, 99.6) | 99.4 (99.1, 99.6) |
| Culture | Sensitivity | 11.5 (9.7, 13.5) | 5.6 (4.3, 7.2) | 76.1 (61.4, 89.5) | 72.7 (57.0, 88.9) | 71.9 (54.6, 88.2) |
| | Specificity | 99.8 (99.6, 99.9) | 99.9 (99.8, 99.9) | 99.9 (99.8, 99.9) | 99.9 (99.8, 99.9) | 99.9 (99.8, 99.9) |
| | Deviance | 5205.7 | 5025.0 | 5017.3 | 5013.5 | 5012.2 |
| | RMSE | 137.0 | 86.1 | 67.5 | 66.9 | 66.6 |

Model 0 –Based on the assumption of conditional independence

Model 1 –Accounts for conditional dependence between radiologist conclusion, CAD4TBv5≥53 and CAD4TBv6≥53 and between Xpert Ultra (excluding trace) and culture among the PTB cases and allows conditional independence between all the diagnostic tests among non-PTB cases

Model 2 –Accounts for conditional dependence between radiologist conclusion, CAD4TBv5≥53 and CAD4TBv6≥53 and between Xpert Ultra (excluding trace) and culture among the PTB cases and conditional dependence between any TB symptom, radiologist conclusion, CAD4TBv5≥53 and CAD4TBv6≥53 among non-PTB cases

Model 3 –Accounts for conditional dependence between all the diagnostic tests except any TB symptom among the PTB cases and conditional dependence between any TB symptom, radiologist conclusion, CAD4TBv5≥53 and CAD4TBv6≥53 among non-PTB cases

Model 4 –Accounts for conditional dependence between all the diagnostic tests among the PTB cases and conditional dependence between any TB symptom, radiologist conclusion, CAD4TBv5≥53 and CAD4TBv6≥53 among non-PTB cases

RMSE–Root mean square error. This is calculated as the square root of the sum of squared differences between the observed frequencies and the predicted frequencies. It shows how good the model is in explaining the variability in the data (smaller is better)

CrI–Credible Intervals, ‡ - Any chest X-ray abnormality, † - Excluding trace

Note: The estimates presented in the table are percentages

dependence between radiologist conclusion (any chest X-ray abnormality), CAD4TBv5≥53 and CAD4TBv6≥53, and between Xpert Ultra (excluding trace) and culture among the PTB cases and continues to assume conditional independence between all the diagnostic tests among non-PTB cases (Model 1) did not resolve the problem. However, allowing for conditional dependence also between any TB symptom, radiologist conclusion, CAD4TBv5≥53 and CAD4TBv6≥53 among non-PTB cases (Model 2) improved the fit and produced very different, more realistic estimates.

The model that accounts for conditional dependence between all the diagnostic tests except any TB symptom among the PTB cases and conditional dependence between any TB symptom, radiologist conclusion, CAD4TBv5≥53 and CAD4TBv6≥53 among non-PTB cases (Model 3) yielded plausible estimates with a deviance of 5013.5 and RMSE of 66.9. Further, the model that accounts for conditional dependence between all the diagnostic tests among the PTB cases and conditional dependence between any TB symptom, radiologist conclusion, CAD4TBv5≥53 and CAD4TBv6≥53 among non-PTB cases (Model 4) produced similar findings to Model 3 with a deviance of 5012.2 and RMSE of 66.6. Although Model 3 and Model 4 had lower deviance, the reduction in deviance does not outweigh the underlying complexity in

these models compared to Model 2. Model 2 is simpler and able to explain the data as good as Model 3 and Model 4.

Based on Model 2, the overall PTB prevalence was 1.1% (95% CrI: 0.7, 1.5) and the overall sensitivity of Xpert Ultra (excluding trace) and culture were 60.0% (95% CrI: 46.9, 72.6) and 76.1% (95% CrI: 61.4, 89.5) respectively. We extended Model 2 to account for the effects of measured covariates (age, sex and HIV status) on PTB prevalence and diagnostic test accuracy. The results are presented in Tables 4 & 5. The deviance from this model was 1656.8 demonstrating that it explains the variability in the data much better than the model without covariates. We repeated the analysis using the same priors for the unknown parameters but with any chest X-ray abnormality replaced by chest X-ray abnormality suggestive of active TB (S10–S14 Tables in *S1 File*). The findings from the two analyses are similar. Nonetheless, the model with any chest X-ray abnormality replaced by chest X-ray abnormality suggestive of active TB explains the data much better with a deviance of 1358.5 (S15 Table in *S1* File).

Table 4 presents the estimates of PTB prevalence and diagnostic test accuracy adjusted for HIV status, sex and age. The overall estimates are standardized by age-, sex- and HIV-specific proportions to the full study population. Compared to the unadjusted model, the age-, sex- and HIV-standardized sensitivity of any TB symptom and culture dropped slightly but increased for radiologist conclusion, CAD4TBv5≥53, CAD4TBv6≥53 and Xpert Ultra (excluding trace). The estimates of specificity for all the diagnostic tests varied slightly. Consequently, the age-, sex- and HIV-standardized PTB prevalence became 0.9 (95% CrI: 0.6, 1.3).

The HIV-stratified estimates are standardized by sex- and age-specific proportions. The age- and sex-standardized PTB prevalence were higher among the HIV+. Compared to the HIV-, the age- and sex-standardized sensitivity (specificity) of any TB symptom, radiologist conclusion, CAD4TBv5≥53 and CAD4TBv6≥53 were higher (lower) among the HIV+. The

**Table 4. Posterior median and 95% credible intervals (95% CrI) of PTB prevalence and diagnostic test sensitivity and specificity for Vukuzazi dataset adjusted for HIV status, sex and age, by HIV status adjusted for age and sex and by sex, adjusted for HIV status and age.**

| | | Overall | HIV+ | HIV- | Male | Female |
|---|---|---|---|---|---|---|
| | N | 4960 (100%) | 1496 (30.2%) | 3464 (69.8%) | 1813 (36.6%) | 3147 (63.4%) |
| Test | Parameter | Median (95% CrI) | Median (95% CrI) | Median (95% CrI) | Median (95% CrI) | Median (95% CrI) |
| | Prevalence | 0.9 (0.6, 1.3) | 1.3 (0.8, 1.9) | 0.8 (0.5, 1.2) | 1.2 (0.8, 1.8) | 0.8 (0.5, 1.2) |
| Any TB | Sensitivity | 26.7 (16.6, 38.6) | 26.8 (14.2, 43.4) | 26.4 (15.5 39.4) | 25.3 (13.0, 40.3) | 27.3 (15.6, 41.4) |
| symptom | Specificity | 83.0 (81.9, 84.0) | 83.5 (81.5, 85.3) | 82.7 (81.5, 84.0) | 83.3 (81.5, 85.0) | 82.8 (81.4, 84.1) |
| Radiologist | Sensitivity | 84.4 (74.0, 91.9) | 88.6 (75.0, 96.1) | 82.8 (70.9, 91.3) | 87.1 (74.4, 95.1) | 83.1 (70.7, 91.5) |
| conclusion‡ | Specificity | 66.7 (65.3, 68.0) | 59.7 (57.2, 62.3) | 69.7 (68.1, 71.2) | 65.8 (63.6, 68.0) | 67.1 (65.4, 68.8) |
| CAD4TBv5≥53 | Sensitivity | 80.4 (68.4, 89.7) | 83.3 (67.8, 93.6) | 79.5 (65.7, 89.8) | 84.7 (70.2, 94.0) | 78.3 (64.6, 89.1) |
| | Specificity | 80.0 (78.9, 81.2) | 78.9 (76.8, 80.9) | 80.6 (79.2, 81.9) | 73.3 (71.2, 75.3) | 84.0 (82.7, 85.2) |
| CAD4TBv6≥53 | Sensitivity | 81.1 (69.2, 90.1) | 82.8 (67.0, 93.1) | 80.7 (67.6, 90.4) | 82.5 (67.2, 92.7) | 80.7 (67.1, 90.5) |
| | Specificity | 82.5 (81.4, 83.6) | 79.5 (77.4, 81.5) | 83.8 (82.6, 85.0) | 76.7 (74.7, 78.6) | 85.9 (84.6, 87.0) |
| Xpert Ultra† | Sensitivity | 62.2 (48.7, 74.4) | 59.9 (41.6, 76.9) | 63.1 (48.1, 76.7) | 70.1 (52.7, 84.1) | 57.7 (42.4, 72.1) |
| | Specificity | 99.4 (99.1, 99.6) | 99.4 (99.1, 99.6) | 99.4 (99.1, 99.6) | 99.4 (99.1, 99.6) | 99.4 (99.1, 99.6) |
| Culture | Sensitivity | 75.9 (61.9, 89.2) | 74.6 (54.6, 91.3) | 76.6 (62.1, 89.6) | 76.2 (58.0, 91.6) | 76.0 (60.5, 89.6) |
| | Specificity | 99.8 (99.7, 99.9) | 99.8 (99.7, 99.9) | 99.8 (99.7, 99.9) | 99.8 (99.7, 99.9) | 99.8 (99.7, 99.9) |

CrI–Credible Intervals

‡ - Any chest X-ray abnormality

† - Excluding trace

Note: The estimates presented in the table are percentages

**Table 5. Age, sex and HIV adjusted posterior median and 95% credible intervals (95% CrI) of PTB prevalence and diagnostic test sensitivity and specificity for Vukuzazi dataset presented by age groups.**

|  |  | 15–29 years | 30–49 years | 50–69 years | ≥70 years |
|---|---|---|---|---|---|
|  | N | 1156 (23.3%) | 1291 (26.0%) | 1723 (34.7%) | 790 (15.9%) |
| Test | Parameter | Median (95% CrI) | Median (95% CrI) | Median (95% CrI) | Median (95% CrI) |
|  | Prevalence | 0.8 (0.5, 1.2) | 1.0 (0.6, 1.6) | 1.0 (0.6, 1.6) | 0.7 (0.4, 1.2) |
| Any TB | Sensitivity | 27.5 (16.0, 42.2) | 22.7 (9.7, 41.5) | 27.8 (13.7, 46.4) | 26.5 (10.0, 48.7) |
| symptom | Specificity | 81.5 (79.1, 83.7) | 83.7 (81.6, 85.7) | 83.5 (81.7, 85.2) | 82.9 (80.1, 85.4) |
| Radiologist | Sensitivity | 81.2 (68.0, 90.5) | 84.1 (67.5, 94.0) | 88.4 (73.7, 96.3) | 83.7 (63.4, 95.1) |
| conclusion‡ | Specificity | 86.8 (84.8, 88.7) | 65.2 (62.5, 67.8) | 59.4 (57.1, 61.7) | 55.5 (51.9, 59.0) |
| CAD4TBv5≥53 | Sensitivity | 79.4 (65.2, 89.6) | 82.2 (63.7, 94.0) | 81.4 (63.3, 93.0) | 79.8 (58.8, 93.6) |
|  | Specificity | 93.7 (92.2, 95.0) | 82.4 (80.2, 84.4) | 76.6 (74.6, 78.6) | 63.9 (60.6, 67.1) |
| CAD4TBv6≥53 | Sensitivity | 79.7 (65.6, 90.1) | 81.0 (61.7, 93.4) | 83.5 (66.6, 93.8) | 81.2 (60.0, 94.0) |
|  | Specificity | 95.8 (94.5, 96.8) | 83.7 (81.6, 85.7) | 78.6 (76.6, 80.5) | 69.7 (66.7, 72.7) |
| Xpert Ultra† | Sensitivity | 62.7 (46.7, 76.5) | 63.7 (43.0, 82.0) | 63.8 (44.1, 80.6) | 56.5 (33.3, 78.2) |
|  | Specificity | 99.4 (99.1, 99.6) | 99.4 (99.1, 99.6) | 99.4 (99.1, 99.6) | 99.4 (99.1, 99.6) |
| Culture | Sensitivity | 77.8 (62.2, 90.4) | 77.3 (54.4, 93.4) | 72.0 (50.9, 90.5) | 82.6 (61.4, 94.8) |
|  | Specificity | 99.8 (99.7, 99.9) | 99.8 (99.7, 99.9) | 99.8 (99.7, 99.9) | 99.8 (99.7, 99.9) |

CrI–Credible Intervals

‡ - Any chest X-ray abnormality

† - Excluding trace

Note: The estimates presented in the table are percentages

age- and sex-standardized sensitivity for Xpert Ultra (excluding trace) and culture were higher among the HIV-.

The sex-stratified estimates were standardized by age- and HIV-specific proportions. The age- and HIV-standardized PTB prevalence was higher among males. The age- and HIV-standardized estimates of sensitivity (specificity) for radiologist conclusion, CAD4TBv5≥53 and CAD4TBv6≥53 were higher (lower) among males. Compared to females, the estimates of sensitivity for Xpert Ultra (excluding trace) and culture were higher among the males.

Table 5 presents the sex- and HIV-standardized estimates of PTB prevalence and diagnostic test sensitivity and specificity by age groups. The sex- and HIV-standardized estimate of PTB prevalence is higher among 30–69 year-old. The specificity of radiologist interpretation (any chest X-ray abnormality), CAD4TBv5≥53 and CAD4TBv6≥53 decrease with increasing age. CAD4TBv6≥53 has a higher specificity compared to CAD4TBv5≥53 across all the age groups. Culture (Xpert Ultra (excluding trace)) has higher (lower) sensitivity among individuals aged ≥70 years.

The estimates of PTB prevalence and diagnostic tests sensitivity and specificity stratified by HIV, sex and age are presented in (S8, S9 Tables in S1 File).

With the same priors for the parameters in Model 2, the analysis of the set of six diagnostic tests with the "trace" category included among the Xpert Ultra positives yielded an overall PTB prevalence that was similar to that obtained with the analysis excluding the "trace" category. Xpert Ultra (including trace) had a higher sensitivity estimate while culture had a lower sensitivity estimate in the analysis of the set of six diagnostic tests with the "trace" category included among the Xpert Ultra positives compared to the analysis that excluded the "trace" category (S15 Table in S1 File). Based on deviance, the analysis of the set of six diagnostic tests with the "trace" category excluded from the Xpert Ultra positive explains the variability in the data much better than the analysis that includes the "trace" category.

## Discussion and conclusion

We proposed an extension of the Bayesian latent class analysis (LCA) approach by Berkvens et al (2006) modelling the probabilities of diagnostic test results conditional on unobserved PTB status and 'earlier' test results in order, using probit regression methods for dependent binary outcomes [15, 17]. Unknown parameters of the sequential regression models were assigned Gaussian priors. Expert opinion was utilized to incorporate potential dependencies and prior knowledge for some parameters. We extended the model to incorporate measured covariate effects.

We applied our approach first to analyze childhood pulmonary TB (CPTB) data that has previously been published [9, 10]. The findings from our proposed model incorporating the expert opinion matched those reported in the previous analyses [10]. The findings from the model that accounts for conditional dependence between all the diagnostic tests but radiography among the children with CPTB and additionally accounts for conditional dependence between TST and radiography among the children without CPTB explained the data much better. The dependence between TST and radiography among children without CPTB may be attributable to other respiratory diseases. The estimates of sensitivity for smear microscopy, Xpert and culture are slightly higher for our proposed model. These findings demonstrate the ability of Model 2 to incorporate all possible sources of variation and dependencies among the diagnostic tests. Hence, weighting appropriately the information from these diagnostic tests may yield better diagnoses. Based on RMSE, this model was able to do better prediction compared to the results based on the expert opinion and those in [10]. Additionally, it performed as good as the model that adjusted for covariates [9].

To the best of our knowledge, our secondary analysis of the Vukuzazi dataset is the first application of Bayesian LCA on a community-based active TB case-finding and the first application of probit regression methods for dependent binary data to address diagnostic test dependencies. In our analysis, the model that considered conditional dependence between radiologist conclusion (any chest X-ray abnormality), CAD4TBv5≥53 and CAD4TBv6≥53 as well as between Xpert Ultra (excluding trace) and culture among the PTB cases and also conditional dependence between any TB symptom, radiologist conclusion (any chest X-ray abnormality), CAD4TBv5≥53 and CAD4TBv6≥53 among non-PTB cases explains well the variability in the data, was simpler than the competing models and produced plausible estimates. Upon adjusting for the measured covariate effects (age, sex and HIV status), the accuracy of the model improved. The age-, sex- and HIV-standardized estimate of overall PTB prevalence was close to the overall population estimate of 0.8% reported in [22, 32, 33]. This larger estimate of PTB prevalence from our model may be explained by the fact that this was a high-risk group for PTB based on the selection to receive microbiological testing if they reported any cardinal TB symptom and/or had an abnormal chest X-ray finding based on CAD4TBv5 score above a predefined threshold. This estimate is nonetheless lower than 1.5% based on a composite reference standard of Xpert Ultra (excluding trace) and culture. In keeping with the findings reported in [32, 33], our analysis produced higher PTB prevalence among males compared to females, among HIV+ compared to HIV-, and among 30–69 year-old compared to 15–29 year-old and ≥70 year-old.

The overall estimate of sensitivity for any TB symptom was low in the community-based multi-morbidity screening conducted in the rural district of uMkhanyakude in northern KwaZulu Natal, South Africa. This agrees with the findings by Govender et al. 2021 [22, 23] who reported that 78% of culture-positive cases did not report symptoms. However, it is higher than the 58% reported in the 2018 South Africa national TB prevalence survey [32, 33]. It has been shown that 49.4% (Inter quartile range: 38.8%– 52.4%) of bacteriologically confirmed

prevalent infectious TB cases are asymptomatic in Africa [11]. Data on this state of TB disease is limited, but is estimated to last around 3–8 months (with three countries in Asia where it lasted >1 year) and represents 27% - 63% of the time as prevalent cases [34]. Hence, this may explain the high proportion of subclinical TB cases estimated in our study that has been shown elsewhere to be as high as 80% [11]. Additionally, it has been argued elsewhere that TB is still highly stigmatized in rural areas [35]. Consequently, symptomatic people who are well aware of the consequences of being diagnosed with TB in the community might have chosen not to report the presence of TB-compatible symptoms. The national survey incorporated the urban and rural areas while the community-based multi-morbidity survey was conducted in a rural district in northern Kwa-Zulu Natal. Another possible explanation for the high proportion of subclinical TB in this study is linked to the use of a low triaging threshold (score of 25) for CAD4TBv5 that triggered sputum testing from a majority of the participants leading to a higher proportion of bacteriologically confirmed asymptomatic TB cases [12]. The specificity of 83% for any TB symptom was low for a community-based TB survey. The high false positive rate is potentially due to reported symptoms akin to other respiratory diseases.

The sensitivity estimate of any chest X-ray abnormality was consistent with what has been reported based on an imperfect reference standard defined as a combination of microbiological tests in TB prevalence surveys [11]. Our estimate was slightly higher than the 80.8% reported in the analysis of the same data that defined the reference standard as a combination of culture and Xpert Ultra (including trace) [12]. Analysis based on a composite reference standard (CRS) defining a case as bacteriologically confirmed if culture positive or Xpert Ultra (including trace) positive with chest X-ray abnormality suggestive of active TB and without current or previous TB show that 98% of bacteriologically confirmed TB cases had any chest X-ray abnormality [33]. The upward biased sensitivity estimate of any chest X-ray abnormality reported in the South Africa national TB prevalence survey is attributed to the definition of the reference standard (Xpert Ultra (including trace) positives were qualified if chest X-ray was suggestive of active TB) and the fact that this imperfect reference standard was considered perfect. The specificity estimate of any chest X-ray abnormality was consistent with 66.9% reported in [12]. The low specificity may be attributed to the effect of radiological changes in the lungs due to other respiratory diseases e.g. bacterial pneumonia, chronic obstructive pulmonary disease or past TB that results in a high false positive rate. In this analysis, LCA was able to utilize information from all the diagnostic tests to estimate the likelihood of TB. This allowed correct classification of individuals. Consequently, it corrected the underestimation of sensitivity and overestimation of specificity. Thus mitigating the reference standard bias to yield plausible estimates. Our analysis that replaced any chest X-ray abnormality with chest X-ray abnormality suggestive of active TB yielded plausible estimates of sensitivity and specificity for chest X-ray abnormality suggestive of active TB. The findings are consistent with the findings reported by Qin et al. 2021 who reported sensitivity and specificity estimates of 38.9% and 88.9%, respectively, based on Xpert MTB/RIF as the reference standard [36].

CAD4TB versions 5 and 6 were both dichotomized at a threshold score of 53 based on LCA. CAD4TBv5$\geq$53 and CAD4TBv6$\geq$53 had acceptable estimates of overall sensitivity and specificity. We established that any chest X-ray abnormality, CAD4TBv5$\geq$53 and CAD4TBv6$\geq$53 have better sensitivity and specificity among individuals aged <30 years.

Although considered more accurate, Xpert Ultra based on DNA of *Mycobacterium tuberculosis* has obvious limitations including the inability to distinguish between dead DNA resulting from past TB infection and live DNA from an active TB. Additionally, the TB bacilli limit of detection (LOD) of $\approx$15.6 CFU/ml for Xpert Ultra results in missed TB cases with a bacillary load that is lower than the LOD. These limitations, among others, render Xpert Ultra imperfect in terms of its sensitivity and specificity [24]. The overall sensitivity of Xpert Ultra

(excluding trace) in our analysis was lower than can be imagined for a perfect diagnostic test. This agrees with the findings reported in [37] and in two community-based TB screening studies conducted in Zambia and South Africa [38] that used culture as the reference standard. The specificity of Xpert Ultra (excluding trace) from our analysis was higher than the 96.6% reported elsewhere with culture as the reference standard [24].

Though considered the most accurate with LOD $\approx$ 1 to 50 CFU/ml, culture still suffers from imperfect sensitivity that spans 73%– 95% [39]. Thus, the overall sensitivity estimate of 76.0% for culture in our analysis is consistent with this finding and confirms the concerns among the clinical experts proposing a composite reference standard with other diagnostic tests considered to be more accurate. Nonetheless, a CRS of imperfect diagnostic tests remains imperfect. Hence alternative approaches such as LCA can be used to navigate these limitations.

There was evidence of subgroup disparities in the estimates of sensitivity and specificity and the subgroup estimates were highly unstable due to small subgroup sample sizes. With careful use of informative priors, if available, the precision of the estimates can improve. Knowledge of the performance of the most accurate tests can be used to elicit a prior distribution for some of the parameters in the model. Based on this approach and previously published work, we proposed an informative prior for the parameters corresponding to the probability that Xpert Ultra (excluding trace) is negative and for the probability that culture is negative among the true non-PTB cases to be around 97% and 99.9%, respectively [24, 39]. We also proposed a prior for the parameter corresponding to the probability that culture is positive among Xpert negative true PTB cases to be 80% (S2-S5 Tables in S1 File).

In our proposed model we only included the main effects of the 'earlier' diagnostic tests and the main effects of covariates in the regressive probit models. However, our model could handle higher-order test dependence by incorporating higher-order interactions among the diagnostic tests. Although the number of parameters required to be estimated grows exponentially as the number of diagnostic tests increases, close consultation with experts can help mitigate the problem. In the CPTB study, the experts were able to provide useful information that radiology was not dependent on the other diagnostic tests given TB status and that all the diagnostic tests were not dependent among children who did not have CPTB. This expert information, obtained from the earlier publications, helped reduce the number of parameters to estimate in the model. In the Vukuzazi study, the TB experts were able to additionally provide insight into the potential causes of dependence among those without PTB. This helped adjust for this dependence leading to plausible estimates of PTB prevalence and diagnostic test accuracies, particularly the sensitivity of Xpert Ultra (excluding trace) and culture. Therefore, working closely with TB experts and incorporating their assumptions into the model can help fix some of the practical limitations. Notwithstanding this, there is a need to assess the plausibility of the model based on its ability to explain variability in the data. Our analysis revealed that constraining all the tests to be independent among the children without CPTB yielded a less accurate model. Thus, our approach was able to identify and incorporate the often overlooked dependence among non-PTB cases. We were also able to establish that the other diagnostic tests were not dependent on any TB symptom among the true PTB cases in the community-based TB survey that enrolled and microbiologically tested the participants who reported any cardinal TB symptom and/or had abnormal chest X-ray findings. Dependence induced by covariates was handled by including measured covariates in the model to reduce residual dependence. This allows for varying diagnostic test accuracy and PTB prevalence across the subpopulations defined by the covariates [19]. Despite the increasing number of parameters to estimate, their inclusion help improve model identifiability. In our analysis of Vukuzazi data, adjusting for HIV status, sex and age reduced further the residual dependence

thus producing realistic estimates of PTB prevalence, and diagnostic test sensitivity and specificity.

Our proposed model is flexible in allowing possible dependencies between the diagnostic tests, flexible choice of priors for the unknown parameters and incorporation of covariates known to affect diagnostic test accuracy and disease prevalence. It produced realistic estimates of sensitivity, specificity and prevalence under interpretable and more plausible assumptions. The model based on the unrealistic assumption of conditional independence and the model that failed to account for dependence between the diagnostic tests among the true non-PTB cases yielded unrealistic estimates. The model that incorporated sources of dependence among the true PTB cases and true non-PTB cases produced plausible estimates. Therefore, all possible sources of diagnostic test dependencies need to be considered to avoid misleading inferences. Additionally, expert opinion and model parsimony can be used to guide/steer the model choice.

## Supporting information

**S1 Checklist. Inclusivity in global research.**
(DOCX)

**S1 File.**
(PDF)

## Author Contributions

**Conceptualization:** Alfred Kipyegon Keter, Lutgarde Lynen, Alastair Van Heerden, Klaus Reither, Els Goetghebeur, Bart K. M. Jacobs.

**Data curation:** Emily Wong.

**Formal analysis:** Alfred Kipyegon Keter, Els Goetghebeur, Bart K. M. Jacobs.

**Funding acquisition:** Lutgarde Lynen, Alastair Van Heerden, Klaus Reither.

**Methodology:** Alfred Kipyegon Keter, Lutgarde Lynen, Alastair Van Heerden, Emily Wong, Klaus Reither, Els Goetghebeur, Bart K. M. Jacobs.

**Resources:** Lutgarde Lynen, Els Goetghebeur, Bart K. M. Jacobs.

**Software:** Alfred Kipyegon Keter, Bart K. M. Jacobs.

**Supervision:** Lutgarde Lynen, Alastair Van Heerden, Els Goetghebeur, Bart K. M. Jacobs.

**Validation:** Lutgarde Lynen, Alastair Van Heerden, Emily Wong, Klaus Reither, Els Goetghebeur, Bart K. M. Jacobs.

**Visualization:** Lutgarde Lynen, Alastair Van Heerden, Emily Wong, Klaus Reither, Els Goetghebeur, Bart K. M. Jacobs.

**Writing – original draft:** Alfred Kipyegon Keter.

**Writing – review & editing:** Alfred Kipyegon Keter, Lutgarde Lynen, Alastair Van Heerden, Emily Wong, Klaus Reither, Els Goetghebeur, Bart K. M. Jacobs.

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
