## [Decision Letter · Decision Letter 0]

15 Feb 2023

Subject: Manuscript submission

Evaluation of tuberculosis diagnostic test accuracy using Bayesian latent class analysis in the presence of conditional dependence between the diagnostic tests used in a community-based tuberculosis screening study

PONE-D-22-34380

Dear Dr. Keter,

We’re pleased to inform you that your manuscript has been judged scientifically suitable for publication and will be formally accepted for publication once it meets all outstanding technical requirements.

Kind regards,

Kamal Kishore Chopra, MBBS, MD

Academic Editor

PLOS ONE

Reviewers' comments:

Reviewer's Responses to Questions

**Comments to the Author**

1. Is the manuscript technically sound, and do the data support the conclusions?

Reviewer #1: Yes

Reviewer #2: Yes

2. Has the statistical analysis been performed appropriately and rigorously? 

Reviewer #1: Yes

Reviewer #2: Yes

3. Have the authors made all data underlying the findings in their manuscript fully available?

Reviewer #1: Yes

Reviewer #2: Yes

4. Is the manuscript presented in an intelligible fashion and written in standard English?

Reviewer #1: Yes

Reviewer #2: Yes

5. Review Comments to the Author

Reviewer #1: Authors have chosen a relevant topic. Current manuscript is technically sound and relevant to the subject concerned. This manuscript can be considered for publication as it enhances the knowledge of the readers of this esteemed journal.

Reviewer #2: Authors of the manuscript have chosen a very pertinent topic keeping in mind the lack of perfect reference standard in the diagnostic of pulmonary tuberculosis. Theoretically, this can be overcome by utilizing appropriate statistical models.

Authors verify their approach by utilizing published data of children suspected of pulmonary TB and also data from a community-based survey conducted South Africa. The current manuscript is an extension of the Bayesian latent class analysis approach by Berkvens et al. If utilized properly, this modelling approach can give reasonable and easily interpretable estimate of pulmonary tuberculosis.

6. PLOS authors have the option to publish the peer review history of their article (what does this mean?). If published, this will include your full peer review and any attached files.

Reviewer #1: No

Reviewer #2: No

---

## [Editor Report · Acceptance letter]

22 Feb 2023

PONE-D-22-34380 

Evaluation of tuberculosis diagnostic test accuracy using Bayesian latent class analysis in the presence of conditional dependence between the diagnostic tests used in a community-based tuberculosis screening study 

Dear Dr. Keter:

I'm pleased to inform you that your manuscript has been deemed suitable for publication in PLOS ONE. Congratulations! Your manuscript is now with our production department. 

Kind regards, 

on behalf of

Dr. Kamal Kishore Chopra 

Academic Editor

PLOS ONE